# *Lingyuanfructus*: The First Fossil Angiosperm with Naked Seeds

**DOI:** 10.3390/life15121827

**Published:** 2025-11-28

**Authors:** Xin Wang

**Affiliations:** State Key Laboratory of Palaeobiology and Stratigraphy, Nanjing Institute of Geology and Palaeontology and CAS Center for Excellence in Life and Paleoenvironment, Chinese Academy of Sciences, Nanjing 210008, China; xinwang@nigpas.ac.cn

**Keywords:** angiosperm, gymnosperm, fossil, plant, evolution, Cretaceous, China

## Abstract

Unlike gymnosperms with naked ovules, angiosperms are defined and characterized by their enclosed ovules. According to plant evolution theories, angiosperms should be derived from their gymnospermous ancestors, which have naked ovules. Thus, an assumed transitional plant is expected to have started but not yet completed the enclosing of its ovules; specifically, some of its ovules are enclosed while others are not. This unusual expectation is, although rational, paradoxical: If this is so, is the plant a gymnosperm or an angiosperm? To date, such an expectation has never been met by any fossil evidence. The lack of favorable evidence makes the above expectation speculative and leaves evolutionary theorists vulnerable to attacks from their opponents. Here, I report a fossil plant, *Lingyuanfructus hibrida* gen. et sp. nov., from the Yixian Formation (Lower Cretaceous) of Liaoning, China, that meets this expectation. With young seeds both naked and enclosed in a single specimen, *Lingyuanfructus* defies any placement among seed plants and blurs the otherwise distinct boundary between angiosperms and gymnosperms, consolidating the foundation for evolutionary theory.

## 1. Introduction

The origin of angiosperms has been a focus of botanical debate for decades [1,2,3,4,5,6,7,8,9,10,11,12,13,14,15,16,17] because it closely hinges on a robust natural angiosperm system, which covers more than 300,000 species (more than 90% species diversity of land plants) that constitute a major portion of the ecosystem in which humans live. A basic unit of an angiosperm gynoecium is a carpel, which is a structure that usually completely encloses one or multiple ovules that mature into seeds during later development. The origin of carpels is a crucial research topic that is closely hinged on the origin of angiosperms. Naturally, it is a focus of debate in botany. Historically, there were two schools on the nature of carpels. A dominating school was led by Arber and Parkin [18], who interpreted a carpel as a derivative of a so-called megasporophyll that bears ovules along its margins. There was another minor school led by Meeuse [19] and Melville [20,21]. The major difference between these two schools is whether a carpel is foliar or axial in nature. Both sides have shown evidence favoring their points of view and refuse to examine the rationale underlying their opponents’ claims. Although there were various pieces of anatomical, morphological, and palaeobotanical evidence favoring the dominating school led by Arber and Parkin [2,22,23], there are also enough flaws, data manipulations, and misinterpretations inflicting these arguments [24]. Various independent palaeobotanical and botanical developments seem to converge to the conclusion that a carpel is a composite organ that includes both foliar and axial parts [24,25,26,27,28,29,30,31,32]. It appears that this “reconciling” interpretation has integrated the evidence of two formerly confronting schools. Theoretically, angiosperms with enclosed ovules should be derived from their gymnospermous ancestors that have naked ovules, and a plant that is supposedly transitional between angiosperms and gymnosperms should have started but not yet completed its ovule-enclosing process, which means that both enclosed and naked ovules occur in this single plant. Although such an expected scenario has been honored by a tree of *Michelia *(Magnoliaceae) [26], and ovules in *Amborella*, *Illicium*, and *Reseda* are not physically fully enclosed, especially in their early stages [27,33,34], such an expected transient evolution snapshot has hitherto been missing in the fossil world, despite the Yixian Formation (Barremian–Aptian, Lower Cretaceous), which is frequently taken as a promising bonanza for solving the problem of angiosperm origin, has yielded diverse angiosperms including *Chaoyangia liangii*, *Archaefructus liaoning*, *A. sinensis*, *Sinocarpus decussatus*, *Nothodichocaprum lingyuanensis*, *Neofructus lingyuanensis*, *Varifructus lingyuanensis*, etc. [1,2,3,4,5,6,35,36,37,38] and numerous mesofossil flowers have been reported from the Cretaceous of Europe and North America [39]. The current fossil record leaves an apparent lacuna in our knowledge of early angiosperm evolution. Here, I report a new angiosperm, *Lingyuanfructus hibrida* gen. et sp. nov, from the Yixian Formation of Lingyuan, Liaoning, China. Notably, although *Lingyuanfructus* clearly has enclosed ovules/seeds, which indicates that it is an angiosperm, another physically connected fruit pair on the same specimen has both enclosed and naked ovules/seeds, a theoretically expected scenario. Although such a rarely seen morphological assemblage defies any decent placement of *Lingyuanfructus* in angiosperms or gymnosperms, it does fill a formerly enormous morphological gap between angiosperms and gymnosperms. Although its Early Cretaceous age downplays its potential implications for angiosperm origin, *Lingyuanfructus* blurs the formerly distinct boundary between gymnosperms and angiosperms, underscoring the gradualism of plant evolution.

## 2. Materials and Methods

The Yixian Formation lies unconformably over the Tuchengzi Formation and is overlain by the Jiufotang Formation in western Liaoning, China [40,41,42,43,44,45,46,47]. It is mainly composed of dark gray to black, gray and purplish-red andesites; basalts; grayish-green, grayish-yellow, and dark gray to black tuff, tuffaceous sandstone; grit and sandy shale; silty mudstone; shaly tuffaceous silty mudstone and siltstone, sandstone, and basal tuffaceous conglomerates [40,48]. The formation is widely distributed in western Liaoning, eastern Inner Mongolia, northern Hebei, and southern Mongolia. The formation is especially fossiliferous, having yielded abundant charophytes, plant fossils, conchostracans, ostracods, insects, gastropods, bivalves, fishes, amphibians, reptiles, birds, and mammals [48]. Previous radiometric and palaeomagnetic dating of the Yixian Formation indicates a Barremian–Aptian age, and recent ^40^Ar/^39^Ar dating indicates that the age of the Yixian Formation is between 129.7 ± 0.5 and 122.1 ± 0.3 Ma [49]. Although there used to be some controversies over the age of the Yixian Formation [2,41,50,51], it appears that a general consensus on the age of the Yixian Formation has been reached since Dilcher et al. have accepted an Early Cretaceous age for the formation [52].

The flora of the Yixian Formation includes Bryophytes, Equisetales, Lycophytes, Filicales, Cycadales, Ginkgoales, Czekanowskiales, Coniferales, Bennettitales, Gnetales, Caytoniales, and Angiosperms, including over 151 species in 75 genera [4,5,6,35,40,53,54,55]. The Yixian Formation is famous for its early angiosperms, including *Chaoyangia* [1], *Archaefructus* [2,3,35,56,57], *Sinocarpus* [4,5], *Callianthus*, *Nothodichocarpum*, *Neofructus*, *Lingyuananthus*, and *Archaebuda* [6,58].

The specimen studied here was collected from an outcrop of the Yixian Formation near Dawangzhangzi Village, Lingyuan, Liaoning, China (41°15′ N, 119°15′ E), including two facing parts of gray siltstone, preserved as a compression with little coalified residue. The general morphology of the specimen was photographed via a Nikon D300 digital camera. The morphological details were observed and photographed via a Nikon SMZ1500 stereomicroscope with a Nikon DS-Fi1 digital camera. For the details of exposed young seeds, SEM was performed via a TESCAN MAIA 3 GMU (TESCAN, Brno, Czech) housed at the Nanjing Institute of Geology and Palaeontology, Nanjing, China. All figures were organized via Photoshop 7.0 for publication.

## 3. Results

*Lingyuanfructus* gen. nov.

**Diagnosis**: Distal portion of plant, including branch and carpels. Branch slender and straight. Flower perianthless, pistillate, including paired carpels. Young seeds multiple per carpel, enclosed, rarely naked, attached to either dorsal or ventral margins.

**Etymology**: *Lingyuan* for the fossil locality near Lingyuan, Liaoning, China (41°15′ N, 119°15′ E); *-fructus* for fruit in Latin.

**Remarks:** Diverse early angiosperms, including *Chaoyangia liangii*, *Archaefructus liaoning*, *A. sinensis*, *Sinocarpus decussatus*, *Nothodichocaprum lingyuanensis*, *Neofructus lingyuanensis*, *Varifructus lingyuanensis*, etc. [1,2,3,4,5,6,35,36,37,38] have been documented in the Yixian Formation. Among them, *Archaefructus* might appear similar to *Lingyuanfructus* in certain aspects. However, a careful examination can easily find the distinctions between these contemporaneous taxa. *Archaefructus* [2,3,35,57] has been repeatedly well studied by various scholars. Its differences from *Lingyuanfructus* include the following. (1) The exposed ovules in *Lingyuanfructus* have never been seen in *Archaefructus* and (2) the ovules/seeds are found inserted on the dorsal/abaxial carpel margin in *Archaefructus* [35,57], which is distinct from the ovule/seed insertion on both carpel margins in *Lingyuanfructus*. These distinctions apparently are enough to justify a new genus for *Lingyuanfructus*.

*Lingyuanfructus hibrida* gen. et sp. nov. (Figure 1 and Figure 2)

**Diagnosis**: Fossil including physically connected branch and two pairs of carpels. Branch including at least three internodes. Internode up to 21 mm long and 1.5 mm wide. Each carpel elongated oval-shaped, 9–11 mm long and 2.8–3.7 mm wide, enclosing 10–15 young seeds. Young seeds elongated or oval-shaped, 0.6–2.5 mm long and 0.6–1.4 mm wide. Other than young seeds enclosed in carpels, some young seeds naked. Young seeds attached to adaxial carpel margin. Young seeds 1–2.3 mm long, 1.1–1.5 mm in diameter, orthotropous, unitegmic, sessile. Integument 0.34 mm thick and 1.2 mm long. Nucellus 0.93 mm long and 0.65 mm in diameter, and free from integument except at base.

**Description**: The fossil is a compression, including two facing parts, embedded in yellowish siltstone. The fossil is 8.3 cm long and 3.7 cm wide, including branches and at least two pairs of physically connected carpels (Figure 1a). The branches include at least three internodes, up to 1.7 mm in diameter basally, weakly tapering to the distal (Figure 1a). The internode is up to 21 mm long and 1.5 mm wide (Figure 1a). At the branch terminal, there is a 7 mm long shared stalk that furcates into two approximately-7.5 mm-long stalks (Figure 1a). Each of the separated stalks bears a pair of carpels that are smoothly transitional to the carpels, showing no traces or scars of perianth or other lateral appendages (Figure 1a). Each carpel is elongated oval-shaped, 9–11 mm long and 2.8–3.7 mm wide (Figure 1a,d and Figure 2a). One of the carpels appears to be damaged by some animals (Figure 2a). There are 10–15 young seeds in each carpel, which are attached to both margins of the carpels (Figure 1a,d–f). Each young seed is elongated or oval-shaped, 0.6–2.5 mm long and 0.6–1.4 mm wide (Figure 1d–f and Figure 2a–h). Although most young seeds are enclosed in the carpels, some young seeds are naked and attached to the adaxial margin of the carpels (Figure 1d,e and Figure 2a,d,g). These young seeds (ovules) are 1–2.3 mm long, 1.1–1.5 mm in diameter, orthotropous, unitegmic, and sessile (Figure 1d,e and Figure 2a–h). The integument is 0.34 mm thick and 1.2 mm long, whereas the nucellus is 0.93 mm long and 0.65 mm in diameter and is free from the integument except at the base (Figure 2b,c,f). The micropyle is approximately 0.5 mm wide (Figure 2b,c,f).

**Remarks**: Although with naked young seeds, *Lingyuanfructus* resembles none of the reproductive organs in known fossil or living gymnosperms, strobili, or other reproductive organs, this is not at odds with the enclosed young seeds observed in the carpels of *Lingyuanfructus* (Figure 1d–f and Figure 2a), which point to an angiospermous affinity.

Leaves closely associated (not physically connected) with *Lingyuanfructus* are strap-shaped, and 20 to >44 mm long and 1.3 mm wide, smooth-margined, and parallel-veined, with rare mesh (Figure 1a–c). Currently, their physical connection to the fruits/carpels cannot be ascertained. This is a question to answer in the future.

The term “carpel” is preferred to the term “fruit” since no seed coat expected for a mature seed is seen in the fossil, whereas the integument and micropyle expected for an ovule are seen in the fossil of *Lingyuanfructus hibrida* (Figure 2b,c,f). Since it is difficult to distinguish an ovule from a young seed in *Lingyuanfructus* (actually, also in extant angiosperms), I prefer to use the term “young seed” instead of “ovule”, to be conservative.

Since this is the first case of angiosperms bearing naked ovules/young seeds, such a rarity should be treated with caution, and all alternatives should be eliminated before proceeding further. The fossil we face now is a consequence of a prolonged taphonomic process, diagenesis, and fossilization. To the best of my knowledge, there is no report that the naked young seeds could result from carpel damage, displacement, or other taphonomic processes. Such a lack of case reports forms a strong contrast with the coherent cellular details preserved in *Lingyuanfructus* (Figure 2a–g), which favor the existence of *in situ* ovules in *Lingyuanfructus*. Even if “naked young seeds could result from carpel damage or displacement” in a carpel/fruit pair in *Lingyuanfructus*, the lack of similar observation in another carpel/fruit pair that is physically connected apparently rejects such an interpretation. Considering these two carpel/fruit pairs are just centimeters away and must have undergone the same taphonomy, diagenesis, and fossilization, and should also demonstrate similar ovule displacement. This contrast between two physically connected carpel/fruit pairs indicates that the observed *in situ* ovules in *Lingyuanfructus* are most likely a reflection of its original natural morphology rather than an artifact due to taphonomic process.

The possibility that young seeds of another plant incidentally fall over the carpel of *Lingyuanfructus*, resulting in an artifact that seemed as if *Lingyuanfructus* had naked young seeds, can be easily eliminated for the following reasons. (1) There is no trace of the second plant in the specimen. (2) Normally, ovules or immature seeds of another plant would not leave their mother plant before maturing as seeds, since, if they did, it would lead to no seed-setting and the discontinuation of lineage, and furthermore, the end of the phylogeny. What is seen in Figure 1d and Figure 2a–f are apparently ovules rather than seeds of *Lingyuanfructus*. (3) Two naked young seeds of *Lingyuanfructus* are physically connected to the carpels (Figure 1d and Figure 2a–d,g). (4) The physically connection between the carpel margin and naked young seeds is confirmed by coherent tissue texture with cellular details that is across the border between these two parts (Figure 2e,g). (5) The young seed of another plant could not fall onto the carpels of *Lingyuanfructus* in such a half-in-half-out status (Figure 2d). (6) The probability of young seeds of other plants falling onto the specimen of *Lingyuanfructus* at the exact expected positions and appearing to be *in situ* is as slim as a person being hit by a meteorite. I personally do not think that this could be the case in *Lingyuanfructus*. In short, it is unlikely that the naked ovules in *Lingyuanfructus* are from another plant.

**Horizon**: The Yixian Formation, Barremian–Aptian, Lower Cretaceous.

**Holotype**: PB328298a, PB23898b.

**Etymology**: *hibrida* for chimeric morphology of the fossil spanning gymnosperms and angiosperms.

**Depository**: Nanjing Institute of Geology and Palaeontology, Nanjing, China.

## 4. Discussion

Literally, angiosperms are characterized by “angiospermy”, which implies that their seeds are enclosed. In this way, the seeds are better protected and nourished in angiosperms, and thus have more chances to give rise to more offspring and thus diversify during their evolution. A stricter criterion that ensures an angiospermous affinity is angio-ovuly: ovules enclosed before pollination [6,59]. Since most ovules/young seeds of *Lingyuanfructus* are enclosed in carpels (Figure 1d,e and Figure 2a), it is reasonable to place *Lingyuanfructus* in angiosperms if taking only the enclosed ovules into consideration. However, the intriguing feature of *Lingyuanfructus* is its naked young seeds, which are a characteristic of gymnosperms. Therefore, it is premature to pin down the affinity of *Lingyuanfructus* for the time being. However, this does not reduce the systematic significance of *Lingyuanfructus*; instead, it underscores its significance in plant systematics and evolution.

In addition to enclosed young seeds, an intriguing feature of *Lingyuanfructus* is its naked young seeds attached to the adaxial margin of the carpels (Figure 1a,d,e and Figure 2b–d). The “nakedness” of these young seeds implies either (1) that *Lingyuanfructus* is a gymnosperm, or (2) that *Lingyuanfructus*, if an angiosperm, has not fully completed its transition from gymnospermy to angiospermy, and thus is a stepping stone between gymnosperms and angiosperms, two groups otherwise well-separated in seed plants.

Previously, a tree of extant *Michelia figo* (Magnoliaceae) in the Botanical Garden of the Ruhr University Bochum (Germany) had demonstrated both naked and enclosed ovules in a single plant or even in a single flower [26]. Such a chimeric character assemblage makes the plant a chimera because it has characteristics of both gymnosperms and angiosperms [26]. A similar example has been lacking in fossil angiosperms hitherto. Such a lack appears normal for fossil plants, considering the rarity of fossils. However, considering that many ancient fossil plants must have undergone this then-common process, theoretically, such fossils should be there although not found. Before *Lingyuanfructus*, the above hypothesis was purely speculative. *Lingyuanfructus*, as the first fossil angiosperm with naked ovules/young seeds, corroborates the above hypothesis and turns it from a speculative hypothesis to a cofirmed evolutionary theory.

*Lingyuanfructus*’ orthotropous ovule (Figure 2b,c,g) implies that such an ovule is ancestral, at least not as derived as assumed previously, in angiosperms. This finding is in line with the observation of orthotropous ovules in the outgroup of angiosperms, gymnosperms. Although bitegmic ovules are frequently observed in extant basal angiosperms and their fossil record can be traced back to the Late Carboniferous (Pennsylvanian) [60,61], ovules in most gymnosperms are unitegmic. The occurrence of unitegmic ovules in *Lingyuanfructus*, which is among early angiosperms, is rather expected as it is suggestive of its proximity to gymnosperms and its distance and alienation from crown angiosperms. These two features of *Lingyuanfructus* in tandem point to its position as an intermediate between gymnosperms and angiosperms.

Although the naked young seeds occur on the adaxial margins of the carpels in *Lingyuanfructus *(Figure 1d,e and Figure 2a,g), the young seeds in another physically connected pair of carpels of *Lingyuanfructus* are attached to both carpel margins (Figure 1a,d,e). Such a placentation has been observed in the model plant *Arabidopsis* (Eudicots) and taken as derived in plant systematics. However, it is noteworthy that such a placentation has been observed more than once in early angiosperms. For example, *Neofructus*, another fossil angiosperm from the same fossil locality, also has such a placentation [36]. It is not alone, as ovules of *Archaeanthus* (a mid-Cretaceous angiosperm from Kansas, USA) [22] are also found attached to both margins of the fruits [62]. These consistent and constant observations indicate that the former expectation for marginal placentation in ancestral angiosperms by the traditional theories is spurious and should be updated accordingly.

Notably, the base and stalk of each carpel in *Lingyuanfructus* are smooth, showing no trace or stubs of the perianth or of stamens (Figure 1a,d). This observation suggests that *Lingyuanfructus* has no typical perianth and no stamens, and thus is perianthless and pistillate. This underscores the lack of typical flowers in early angiosperms in the Early Cretaceous, including *Archaefructus* [2,3,35,57], *Sinocarpus* [4,5], *Baicarpus* [63], and *Neofructus* [36] (all from the Yixian Formation). The consensus reached by these early angiosperms seems to suggest that these early angiosperms lack true “flowers” typical of extant angiosperms. The age of true “flowers” is a botanical question that is made complex by the discovery of *Nanjinganthus* with typical flower morphology in the Jurassic [7,64,65].

It is noteworthy that one of the carpels in *Lingyuanfructus* was damaged (Figure 2a). The culprit of such damage appears to be some kind of insect, although this conclusion apparently requires further investigation to confirm. It can be said that early angiosperms in the Yixian Formation have established their ecological interactions with contemporaneous animals, and animals were one of many factors shaping the evolution of angiosperms in the Early Cretaceous. This conclusion is in line with previous conclusions of studies on Jurassic as well as later plants (including angiosperms).

It should be borne in mind that at least some putative angiosperms have been documented in the Early Permian (over 290 Ma ago), whereas the age of *Lingyuanfructus* is much younger (the Early Cretaceous, approximately 125 Ma ago). *Taiyuanostachya* and *Yuzhoua* are two Palaeozoic plants that have enclosed ovules [66,67]. According to the previously discussed and adopted criterion [6,59], *Taiyuanostachya* and *Yuzhoua* are *bona fide* angiosperms. Although these putative Permian angiosperms are not widely accepted yet, pre-Cretaceous angiosperms, including *Schmeissneria* [68,69] and *Nanjinganthus* [7,64,65], have been repeatedly confirmed in the Jurassic. Therefore, the simultaneous occurrence of both gymnospermy and angiospermy in *Lingyuanfructus* apparently cannot be the only transitional taxon between gymnosperms and angiosperms as there are pre-Cretaceous angiosperms, which must have had their own similar transitional precursors. *Lingyuanfructus* is simply one of several (if not many) taxa transitional between gymnosperms and angiosperms. This highlights the issue of the monophyly of angiosperms. Although most botanists believe that angiosperms are monophyletic [70], such a monophyly has never been reliably proven and current fossil evidence appears disfavor the monophyly of angiosperms. For example, some fossil angiosperms have gone extinct [71,72]. By definition, the exclusion of any taxon is enough to demolish the monophyly of the group in which it is placed. Therefore, the monophyly of angiosperms assumed by APG is spurious. The later occurrence of *Lingyuanfructus* in the Early Cretaceous further undermines the monophyly of angiosperms. Although compared with the APG system, the “polyphyletic-polychronic-polytopic” theory previously proposed [73] won less favor among botanists, increasingly available evidence seems to lend badly needed support to this otherwise less-favored theory.

It should be borne in mind that *Lingyuanfructus* sheds less light on the precursors of angiosperms since it is not the only taxon intermediate between gymnosperms and angiosperms. This conservativeness is understandable especially when Jurassic angiosperms (including *Schmeissneria* [68,69], *Nanjinganthus* [7,64,65], and *Euanthus* [74]) and claimed Permian angiosperms (including *Taiyuanostachya* [66] and *Yuzhoua* [67]) are considered. *Lingyuanfructus*, reported here, indicates that there was a transition between gymnosperms and angiosperms. This conclusion confirms the assumed gradual evolution between gymnosperms and angiosperms in geological history, dispelling the suspicion cast over gradualism of plant evolution.

## 5. Conclusions

As the first fossil angiosperm with naked ovules, *Lingyuanfructus* bridges the formerly large morphological gap between gymnosperms and angiosperms. This new observation completes the evidence chain of plant evolution and confirms the gradual evolution between two otherwise distinct plant groups, gymnosperms and angiosperms. 

## Figures and Tables

**Figure 1 life-15-01827-f001:**
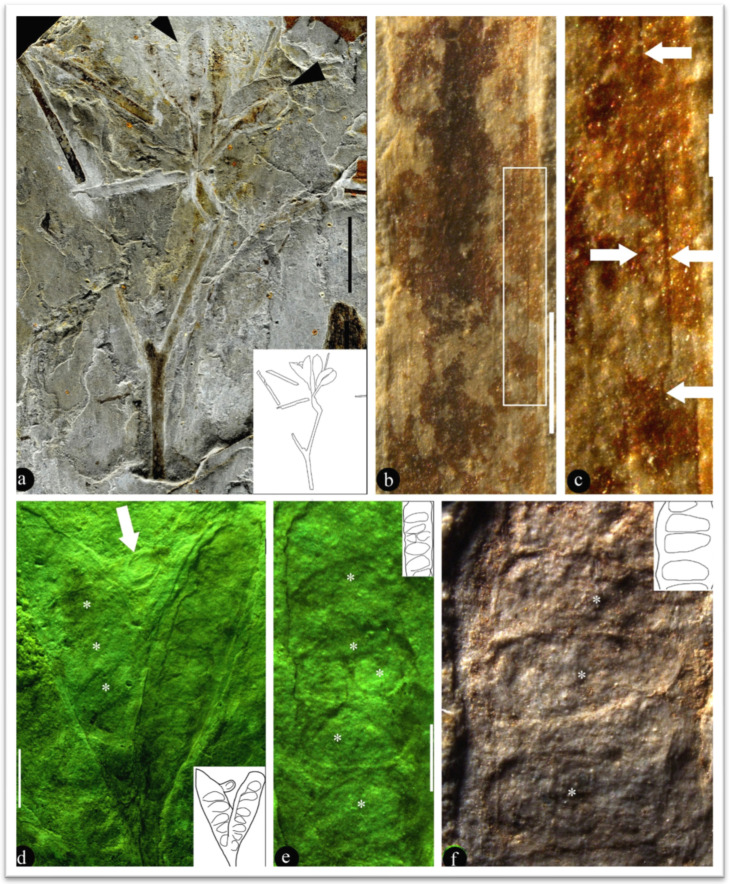
***Lingyuanfructus hibrida* and its details.** PB23898a. (**b**–**f**) Stereomicroscopy. (**a**) Holotype of *Lingyuanfructus*, including physically connected branches and carpels. Refer to the inset sketch. PB23898a. Scale bar = 1 cm. (**b**) Closely associated strap-shaped leaf with smooth margins and parallel venation. Scale bar = 1 mm. (**c**) Parallel venation and a rare mesh (arrows), enlarged from the rectangle in (**b**). Scale bar = 0.2 mm. (**d**) Paired carpels with a young seed (arrow) outside and multiple young seeds (asterisks) inside the connected carpels. Refer to the inset sketch. Note that the young seeds are attached to both fruit margins. Scale bar = 2 mm. (**e**) Detailed view of the young seeds (asterisks) connected to the margin of the left fruit in (**d**). Refer to the inset sketch. Scale bar = 1 mm. (**f**) A row of young seeds (asterisks) inside the right carpel in (**d**). Refer to the inset sketch. Scale bar = 1 mm.

**Figure 2 life-15-01827-f002:**
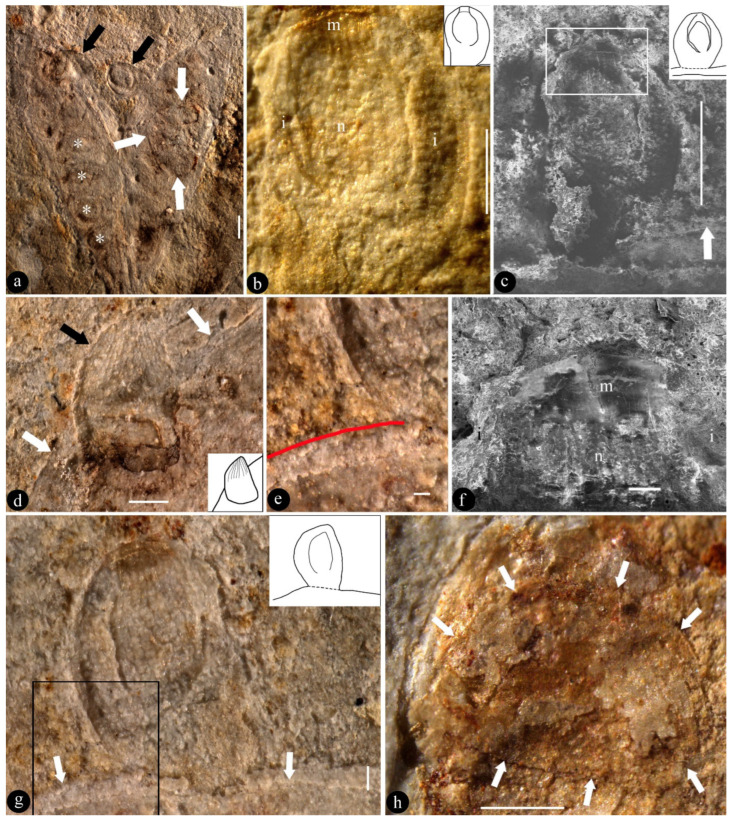
**Details of the naked young seeds of *Lingyuanfructus hibrida*.** (**c**,**f**) SEM; others stereomicroscopy. (**a**) Two carpels with naked young seeds (black arrows) connected to the adaxial carpel margins. Note most young seeds/ovules (asterisks) enclosed in a carpel and the damage mark (white arrows) on the carpel. PB23898b. Scale bar = 1 mm. (**b**) Orthotropous unitegmic ovule outside carpel shown in Figure 1d, showing the nucellus (n), integument (i), and micropyle (m). Refer to the inset sketch. Scale bar = 0.5 mm. (**c**) The same young seed as in (**b**). Note its connection to the horizontal carpel margin (arrow). Refer to the inset sketch. Scale bar = 0.5 mm. (**d**) Detailed view of the naked young seed (black arrow) marked by the left black arrow in (**a**). Note its spatial relationship with the carpel margin (white arrows). Refer to the inset sketch. Scale bar = 0.5 mm. (**e**) Detailed view of the rectangular region in (**g**), showing the border (arrowed in (**g**)) between the young seed (above the red line) and carpel (below the red line) and the coherent cellular texture of these two parts. Scale bar = 0.1 mm. (**f**) Detailed view of rectangular region of the young seed in (**c**), showing nucellus (n), micropyle (m), and integument (i). Scale bar = 0.1 mm. (**g**) Detailed view of the naked young seed marked by the right black arrow in (**a**). Note its spatial relationship with the carpel margin (white arrows). Refer to the inset sketch. Scale bar = 0.2 mm. (**h**) An oval-shaped young seed (arrows) inside a carpel. Scale bar = 0.5 mm.

## Data Availability

All data are published in this paper. The holotype specimen is deposited in Nanjing Institute of Geology and Palaeontology.

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
