# Peer review of "Lingyuanfructus: The First Fossil Angiosperm with Naked Seeds"

_life, 2025, doi:10.3390/life15121827_

Round 1

Reviewer 1 Report

Comments and Suggestions for Authors

Dear author,

The manuscript describes a new fossil linking gymnosperms to angiosperms. It is a fascinating scientific novelty. I suggest improving the introduction to demonstrate its relevance and importance better.

Introduction
Angiosperms have carpels. The introduction is too short, a single paragraph, and does not mention the word "carpel", nor its origin, evolution, and importance for angiosperm evolution. I suggest including more information about the debate over angiosperm origin. This would make a better connection with the discussion and the impact of the discovery of this fossil. 

Discussion

The author cited the "polyphyletic-polychronic-polytopic" theory of angiosperm origin. It is a controversial theory that challenges the idea of a single common ancestry for all angiosperms, the most widely accepted being the APG system. I suggest including one of two sentences to contemplate both theories.

Reviewer 2 Report

Comments and Suggestions for Authors

There are major problems in the ms. 1) taxonomy, 2) morphology/anatomy 3) interpretation

1) As far as I know the group termed as gymno-angiosperms does not exist in current plant taxonomy. So, it is a new taxonomic entity introduced here. There is no statement, or definition what is characteristic for such a group. Taxonomy of a particular plant group is not usually based on only one character. For example angiosperms are not defined by one sole character - closure of the ovule in a carpel, as mentioned in the MS. There are number of diagnostic characters in parallel to the closure of the ovule (e.g. double fertilisation, presence of endosperm, vessels) distinguishing angiosperms from gymnosperms. Those characters do not always occur all in one taxon. So, there is no chimera (Michelia) among Magnoliaceae (the same way absence of vessels in Amborella and Wiinteraceae does not make them chimeric among early angiosperms).

2) Closure of the carpel may be by secretion without the need of post-genital fusion. This is quite a common character among early angiosperms. Secretion itself has low fossilisation potential and can simply be missing in the fossil. However, the fig. 1d (It seems to be the only case of non-closed ovule in the paper) seems to be a taphonomic artefact. More fossil specimens like that are needed for clarification.

3) The manuscript contains number of misleading or at least eccentric statements. E.g. “Lingyuanfructus, as the first fossil plant with both enclosed and naked ovules, does blur the formerly distinct boundary between gymnosperms and angiosperms”. If the fossil shows similar character to Michelia (as the authors state in another part of the ms), which is an angiosperm, the authors should assign their fossil also to angiosperms. Certainly, there is another possibility: the author should try to convince readers, that Michelia is not an angiosperm.

In terms of terminology, ovules usually do not preserve isolated in fossil record. I recommend terming the objects as young seeds.

Smaller items:

Line 84: What does the author mean by terms “of two facing parts”?  Is it a part and counterpart?

Line 186: Michelia figo is an angiosperm. There is no doubt about it. So, you cannot state “Michelia figo is taken as the first gymno-angiosperm”.

Line 249: The author should avoid statement: “angiosperms have been seen in the Early Permian”. This statement is not acceptable and not scientific. This meaning needs a reference or explanation.

Line 258: Referring to his own research the author remains alone in his eccentric ideas.

Line 298: There is no proof for this statement: “As the first fossil gymno-angiosperm..”

Reviewer 3 Report

Comments and Suggestions for Authors

This is an interesting fossil but I do not believe it is a new species, nor does it have any clear indications of angiosperm ancestry. I think this manuscript must be rejected in current form.

My main misgiving is evident from the abstract which has a false premise, that no fossil is known of incompletely closed ovules. In fact Caytonia is such a fossil as long known to have incompletely closed fruits because pollen grains could find their way in. There are many others as well among glossopterids, Leptostrobus, and cheirolepidiacean conifers.

My most serious reservation is that this fossil looks like Archaeofructus from the same formation. How does it differ?  Why a new species?

Another problem is that the supposed leaves do not appear to be attached.

Many very primitive living angiosperms also have incompletely closed carpels including Amborella, Illicium and Drimys, not just Michelia (l. 33).

The ovules are said to be unitegmic (l.127) whereas angiosperms have double integument.

l.71 no need to repeat author should be just Dilcher et al. (2007).

Round 2

Reviewer 2 Report

Comments and Suggestions for Authors

The paper still needs more work.

There are at least major issues to be addressed:

It is always suspect if there is only one specimen on which the taxon is defined. It testifies to the rarity of such a find and creates an impression of uniqueness which is, however, detrimental. So, if more specimens of this kind are known to the author, they should be mentioned.

I searched in the reviewed manuscript, and I did not find any word „taphonomy“. I am quite convinced this fossil require evaluation in terms of taphonomy. As one can imagine the isolated young seeds occurring out of the carpel might easily be a product of taphonomical processes. I personally consider this explanation more plausible than constructing rather wild hypothesis.

So, before I agree with publication of this manuscript, I ask for a paragraph where the author would discuss taphonomy. As mentioned above, it cannot be ruled out that the carpel was partly damaged and opened during the sedimentation and some young seeds ended up outside the fruit. This explanation should be mentioned aside with the author hypothesis about the peculiar fossil “bridging gymnosperms and angiosperms”.

Reviewer 3 Report

Comments and Suggestions for Authors

I have gone over this revised version and still find the paper unacceptable. 

This new text on l.188 is embarassing for a scientific paper "The probability of young seeds of other plants falling onto the specimen of "Lingyuanfructus at the exactly expected positions and appears to be in situ is as slim as a person being hit by a meteorite. I personally do not think that this could be case in Lingyuanfructus. In short, it is impossible for the naked ovules in Lingyuanfructus to be from another plant." Can you just state that it is unlikely?

My two main reservations were lack of evidence that the leaf belongs to the fruit. It is now made plain that there are associated, not attached. My other sticking point was lack of comparison with Archaeofructus, which this new specimen resembles closely. Is it really a new species? I am unconvinced.

Round 3

Reviewer 2 Report

Comments and Suggestions for Authors

I am surprised how offensive is the author. I really do not want to argue anymore with our small Einstein. I have been observing his work for some time now, in which he is increasingly closing himself off from the outside world, especially the opinions of his colleagues. I refuse to discuss it in this way. I leave it up to the editors to decide how they will approach the work.

Reviewer 3 Report

Comments and Suggestions for Authors

OK my major problems have been addressed, but the writing is still awkward in places